# Performance-Based Expressway Asphalt Pavement Structural Surface Layer Modulus Matching Mode Research

**Yonghai He [1], Changyu Pu [1], Peng Xu [2,3,*], Xujia Li [2], Guangqing Yang [2,3] and Huilin Meng [1]**

[1] Hebei Transportation Planning and Design Institute Co., Ltd., Shijiazhuang 050000, China; yonghaihe@126.com (Y.H.); puchangyu_1980@163.com (C.P.); gtsyang@vip.163.com (H.M.)

[2] School of Civil Engineering, Shijiazhuang Tiedao University, Shijiazhuang 050043, China; liuzhuliweiyi@163.com (X.L.); yanggq@stdu.edu.cn (G.Y.)

[3] State Key Laboratory of Mechanical Behavior and System Safety of Traffic Engineering Structures, Shijiazhuang Tiedao University, Shijiazhuang 050043, China

\* Correspondence: sdxplt@163.com

**Abstract:** The current asphalt pavement design theory is a traditional line elastic theory. The voltage resistance recovery modulus is used as the material stiffness parameter in the design, which does not fully consider the significant differences between the road material pull-up modulus. Therefore, to improve the overall stress of the pavement structure, this article started from the perspective of the volume of the structure layer, analyzed the development characteristics of reflected cracks in terms of the modular volume of different surface layers, and studied the size of each layer. It is proposed to match the asphalt pavement layer modulus with the crack development level. Based on structural computing and simulation, the application of different modular matching modes was verified, and support was provided for the design of the pavement structure. The results showed that with the increase of the surface modulus, the stress intensity factor determined by a semi-analytical method showed a nonlinear decrease trend, and the change of stress intensity factor was not obvious when the modulus was greater than 10,000; the surface layer compression mode ratio increased, and the vertical deformation of the road surface and the top surface pressure of the road base slowly increased, especially when the volume of the pressure pull-up ratio was greater than 1.5. In addition, the impact of the constraint between the surface layer on the vertical deformation of the road table decreased with the decrease in the volume of the surface layer.

**Keywords:** asphalt pavement; facial modulus; match mode; reflection crack; stress intensity factor

## 1. Introduction

The asphalt pavement structure of expressways should provide good road performance and durability, as well as high strength and bearing capacity. However, there are a series of problems with the extensive use of asphalt pavement. Due to the repeated action of structural dead weight and vehicle load, the subgrade will suffer from significant deformation, which can cause road problems such as ruts and cracking. The compactness of the subgrade base will let water remain at the top of the subgrade and cause hydrodynamic pressure under vehicle load, which will lead to issues such as scouring and pulping of the subgrade base. The occurrence of early damage to the pavement structure's base will seriously affect the service life of the road and increase the maintenance costs and affect the traffic conditions during the maintenance period, resulting in the reduction of traffic operation efficiency. The structural performance of asphalt pavement is affected by the tensile modulus characteristics of asphalt mixture, the interlayer constraint state of the pavement structure, the performance of the asphalt pavement with high modulus, and the layer-level design. To solve these problems, applying a reasonable mode of surface modulus matching in highway asphalt's pavement structure can effectively alleviate a

series of issues caused by the pavement base. To this end, domestic and foreign scholars have carried out a lot of research.

Chen Shaoxing et al. [1], to accurately obtain the modulus of pavement asphalt mixture, obtained modulus values and their corresponding development rules through four different test methods, including the bending stiffness modulus test, splitting stiffness modulus test, uniaxial compression dynamic modulus test, and FWD (Falling Weight Deflectometer) inverse calculation modulus, and conducted a comparative analysis on the test results of different moduli of asphalt mixtures. The results showed that under the same experimental conditions, the dynamic modulus was the largest, followed by the bending modulus, the splitting modulus, and the inverse modulus. Wang Xiaoyang [2] explored the causes of the differences between the tensile and compressive moduli of asphalt mixtures, and compared and analyzed the influencing factors such as the initial value, critical value, failure turning point, and modulus decay slope of the tensile and compressive moduli in the fatigue process, and then established a decay model of the ensile and compressive moduli of asphalt mixtures in the bending and tensile fatigue test. Pan Qinxue et al. [3] introduced a mechanical calculation method that considered different tension and compression moduli, namely the dual modulus theory, into the mechanical calculation of asphalt pavement and established a mechanical model conforming to the different tension and compression characteristics of road materials and the actual pavement's mechanical properties to study the influence of the properties of different materials' tension and compression moduli on the mechanical response of the pavement structure. Cheng Huilei et al. [4], based on the finite element model under the dual-modulus framework, established a calculation method considering the compression–tension dual modulus and Poisson's ratio of asphalt mixtures in an indirect tensile test and took a dense grade mixture (AC-13) as an example. The difference of the dynamic press–pull dual modulus, press–pull Poisson ratio, single modulus, and single Poisson of the asphalt mixture under different temperatures and loading frequencies was analyzed.

Table 1 above lists the tension/compression modulus ratios of different pavement asphalt mixtures. Although a large number of test data show that the asphalt mixtures have significant anisotropy, it is usually simplified as a homogeneous material in the existing numerical analysis and calculation. Under the action of vehicle load, the stress states at different positions of the pavement structure are different, and there are areas of tension and compression. The compressive modulus used in the current homogeneous model is relatively large compared with other moduli such as tension, so the calculation of the compressive modulus may lead to the calculation result of the internal stress of the pavement structure being too conservative, and the deformation of the structure being underestimated. Therefore, considering the influence of the ensile modulus of the asphalt mixture in the calculation, the mechanical deformation characteristics of the pavement structure can be reflected more accurately to realize the optimal design of the pavement structure.

**Table 1.** Tensile and compression moduli ratios of asphalt mixtures.

| AC-25/20 | AC-13 | SBS Modified Asphalt |
|---|---|---|
| Chen Shaoxing et al. [1] Compression and tension modulus ratio 1.5 | Pan Qinxue et al. [3] Compression and tension modulus ratio 1.5–2 | Wang Xiaoyang [2] Tension and compression modulus ratio 3:5 |
| Xue Yanqing et al. [5] Tension and compression modulus ratio 0.59–0.69 | Cheng Huailei et al. [4] Compression and tension modulus ratio 1.6–4.0 | Yan Xili [9] Compression and tension modulus ratio 3.9–7.8 |
| Ren Yufang et al. [6] Compression and tension modulus ratio 1.31 | Huang Mouying [7] Compression and tension modulus ratio 1.2–1.6 | Lu Songtao et al. [10] Compression and tension modulus ratio 1.2 |
| Huang Mouying [7] Compression and tension modulus ratio 1.3–1.8 | Wang Yangyang [8] Tension and compression modulus ratio 1:2 | Guo Tong et al. [11] Compression and tension modulus ratio 1.6–2 |

High-modulus asphalt was first proposed by French researchers, and usually refers to an asphalt mixture with a complex modulus ≥14,000 MPa under the conditions of 15 °C and 10 Hz. High-modulus asphalt has been widely studied and applied in pavement engineering by scholars because of its good rutting resistance, fatigue resistance, and durability.

High-modulus asphalt mixtures can effectively reduce the thickness of the pavement structure. For example, Carbonneau et al. [12] found that using a high-modulus asphalt mixture as the underlying layer could reduce the total thickness of the entire pavement structure by 25 mm when comparing the mechanical response of two Danish pavements with and without a high-modulus asphalt mixture (Figure 1). Guyot [13] reported the application of a high-modulus asphalt mixture at Sir Sivogul Ramgoolam International Airport and found that the reported thickness could be reduced by 105 m with HMA and the greenhouse gas emissions could be reduced by 13%. A large number of studies in Thailand and France have also shown that using high-modulus asphalt mixtures can reduce the thickness of pavement structures [14].

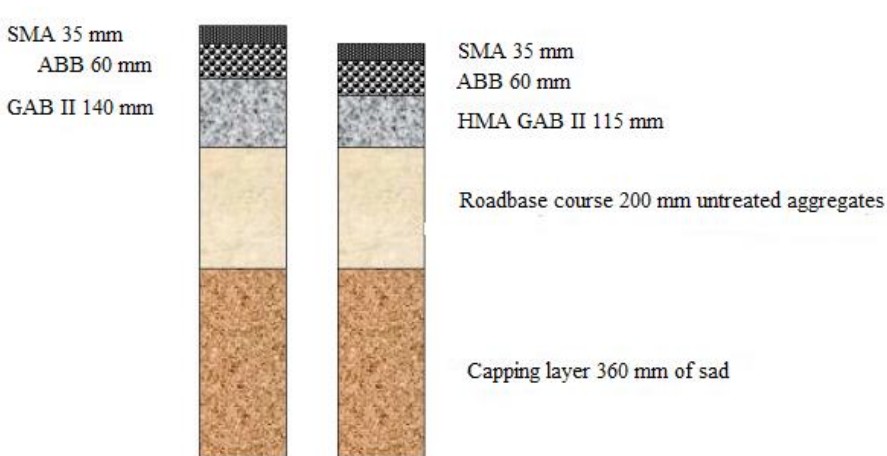

**Figure 1.** Pavement structure.

There is still great controversy about high-modulus asphalt mixtures' horizons. In the typical structure of flexible base long-life asphalt pavement in European and American countries, high-modulus asphalt concrete is mainly used as a rutting-resistant structure layer rather than a fatigue-resistant layer. Leiva-Villacorta et al. analyzed the applicability of high-modulus asphalt concrete as a base [14]. In China, Lu Qingqing and Sun Yana [15] analyzed the mechanical response of high-modulus asphalt mixture located at different layers and found that the pavement performance was the worst when it was located at the upper layer. Wang Kun and Hao Peiwen et al. [16] believe that when high-modulus asphalt mixture is applied to the middle of a road surface, it can effectively improve the high-temperature rutting resistance of the road surface, thus improving its overall performance. Yang Guang and Wang Xudong [17] argued that in semi-rigid base long-life asphalt pavement, the use of high-modulus asphalt concrete in the bottom layer can improve the rutting resistance and fatigue resistance of the pavement [15].

The tensile and compressive modulus of the asphalt mixture has a significant influence on the stress and deformation of the pavement structure. Although a large number of numerical studies on high-modulus of asphalt mixtures have been carried out, the diffusion characteristics of the internal stress in structures with different modulus ratios are still unclear. In addition, for pavement structures during the service period, although some practices show that the increase of the modulus of the surface can improve the performance of the structure to a certain extent, there are still few studies on the quantitative analysis of the degree of improvement of the high-modulus surface on the pavement force and the placement of the high-modulus surface. Therefore, by establishing an asphalt pavement simulation model, this paper explored the matching mode of asphalt pavement surface

modulus when considering the reflection ability of flexural and tensile fatigue cracks and then studied the influence of the matching mode of surface modulus on the deformation of pavement structural layer and the compressive strain of subgrade's top surface.

## 2. Establishment of Asphalt Pavement Simulation Model

### 2.1. Principles of Numerical Simulation

FLAC3D (Free Lossless Audio Codec) uses an explicit Lagrange algorithm and hybrid discrete partition (Fast Lagrange Analysis of Continua), which can accurately simulate the plastic failure and flow of materials. Since there is no need to form a stiffness matrix, a large range of three-dimensional problems can be solved based on small memory space.

The basic idea of the difference method is to use a difference grid to discrete the solution domain, use a difference formula to transform the governing equation of a scientific problem (ordinary differential equation or partial differential equation) into a difference equation, and then combine the initial and boundary conditions to solve the linear algebraic equations. Since this method is intuitive and easy to program, it has been widely used since the 1940s.

### 2.1.1. Finite Difference Equation

The structure is divided into a finite difference grid composed of quadrilateral elements. Within FLAC, each quadrilateral element is divided into two constant-strain triangular elements, as shown in Figure 2.

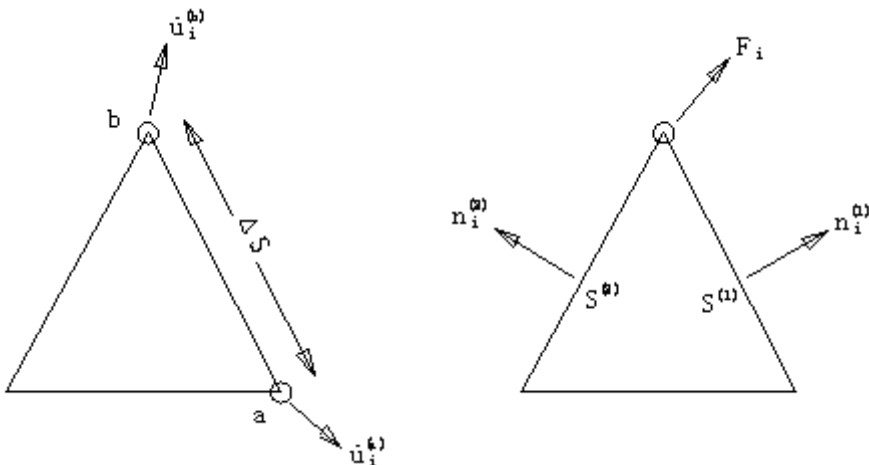

**Figure 2.** Constant strain triangular element.

The triangular difference equation is derived from the general form of the Gaussian divergence theorem, which has the following form:

$$\int_s n_i f ds = \int_A \frac{\partial f}{\partial x_i} dA \tag{1}$$

where, $\int_s$ is the integral around the boundary of a closed surface; $\eta_i$ is the unit normal vector of region $s$; $f$ is a scalar, vector, or tensor; $x_i$ is the coordinate vector; $d_s$ is the incremental arc length; $\int_A$ is the integral of surface area $A$.

The mean value $f$ of the gradient on the area $A$ is defined as follows:

$$\left\langle \frac{\partial f}{\partial x_i} \right\rangle = \frac{1}{A} \int_A \frac{\partial f}{\partial x_i} dA \tag{2}$$

Substitute Equation (2) into Equation (1) to obtain the following:

$$\left\langle \frac{\partial f}{\partial x_i} \right\rangle = \frac{1}{A} \int_s n_i f ds \tag{3}$$

For a triangular subelement, the finite difference form of Equation (3) can become as follows:

$$\left\langle \frac{\partial f}{\partial x_i} \right\rangle = \frac{1}{A} \sum (f) n_i \Delta s \tag{4}$$

where, $\Delta s$ is the length of one side of the triangle, and the right side is the sum of the three sides. $<f>$ is the average of the corresponding edges.

### 2.1.2. Equation of Motion and Displacement Calculation

The equation of motion in the continuum is as follows:

$$\rho \frac{\partial \dot{u}_i}{\partial t} = \frac{\partial \sigma_{ij}}{\partial x_j} + \rho g_i \tag{5}$$

Type in $\rho$ for density, $\dot{u}_i$ for speed; $t$ is time; $x_i$ is the component of the coordinate vector; $g_i$ is the component of gravitational acceleration; and $\sigma_{ij}$ is the component of the stress tensor.

According to Newton's second law of motion, for an object of mass m acting on a force $F(t)$ that varies with time:

$$m \frac{d\dot{u}_i}{dt} = F(t) \tag{6}$$

$$\frac{\partial \dot{u}_i}{\partial t} = \frac{\partial \dot{u}_i^{t+\Delta t/2} - \partial \dot{u}_i^{t-\Delta t/2}}{\Delta t} \tag{7}$$

Substitute Equation (7) into Equation (6) to obtain the following:

$$\partial \dot{u}_i^{t+\Delta t/2} = \partial \dot{u}_i^{t-\Delta t/2} + \frac{F(t)}{m} \Delta t \tag{8}$$

### 2.1.3. Constitutive Relation

The strain rate obtained from the velocity gradient is as follows:

$$\dot{e}_{ij} = \frac{1}{2} \left[ \frac{\partial \dot{u}_i}{\partial x_i} + \frac{\partial \dot{u}_j}{\partial x_i} \right] \tag{9}$$

where, $\dot{e}_{ij}$ is the component of strain rate; $\dot{u}_i$ is the velocity component.

When the strain increment is determined, the stress increment can be calculated from the constitutive equation, and then the total stress can be obtained.

The form of constitutive relation is as follows:

$$\sigma_{ij} := M(\sigma_{ij}, \dot{e}_{ij}, k) \tag{10}$$

where ":" means "by ... ... In place of"; and $k$ represents a historical parameter that may or may not appear, depending on a particular law.

In general, since the correspondence between stress and strain is not unique, the nonlinear constitutive relation is expressed in incremental form.

### 2.1.4. Solution Process

The solution process of the fast Lagrange method is shown in Figure 3.

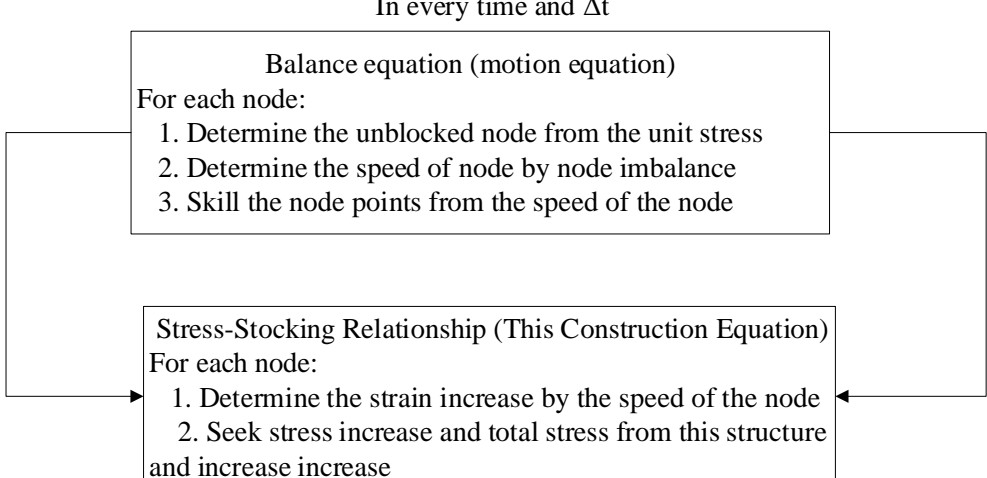

**Figure 3.** Solving process of the fast Lagrange method.

*2.2. Asphalt Pavement Simulation Model*

2.2.1. Basic Assumption

According to the layered system theory, in order to analyze the mechanical and deformation characteristics of asphalt pavement structure by using the finite difference method, the following assumptions were made for the pavement structure:

1. Each layer of asphalt pavement is assumed to be a continuous material. Elastoplastic materials are characterized by elastic modulus, Poisson's ratio, and shear strength;

   The elastic parameters of the material are according to the following relation:

$$G = \frac{E}{2(1 + v)} \tag{11}$$

$$K = \frac{E}{3(1 - 2v)} \tag{12}$$

where, $E$, $G$, and $K$ are elastic modulus, shear modulus, and volume modulus respectively. $v$ stands for Poisson's ratio. As shown in Figures 4 and 5 below, the failure criterion for moll coulomb materials can be expressed as follows:

$$f^s = -\sigma_1 + \sigma_3 \frac{1 + \sin\phi}{1 - \sin\phi} - 2c\sqrt{\frac{1 + \sin\phi}{1 - \sin\phi}} \tag{13}$$

$$f^t = \sigma_3 - \sigma^t \tag{14}$$

where, $C$ is the cohesive force, and $\phi$ is the friction angle. When considering cracks in the material due to tension,

$$f^t < 0 \tag{15}$$

and at this point:

2. The crack is perpendicular to the principal tensile stress $\sigma_3$;
3. Due to instantaneous softening, the tensile strength perpendicular to the crack is 0;
4. Based on the rule of flow and tensile failure envelope update calculation stress, $\lambda^t$ determine the parameters, and the plastic tensile strain increments $\Delta\sigma_1^t = \lambda^t$;

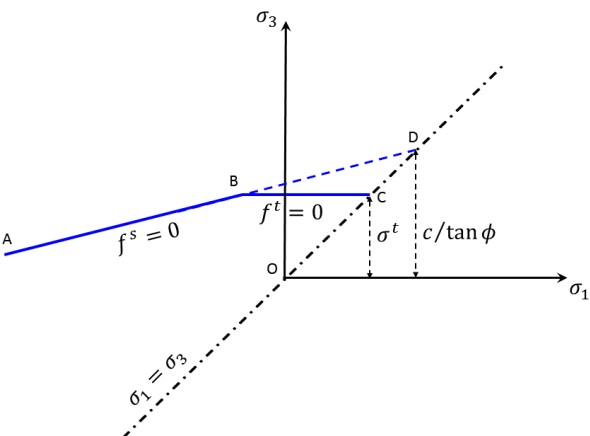

**Figure 4.** MC-T criterion.

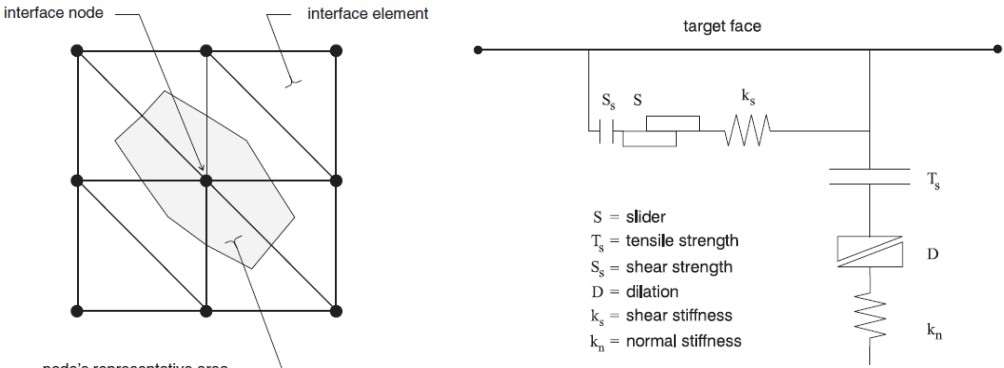

**Figure 5.** Interface interlayer contact model.

There is no emptying between each layer of pavement structure, and the interlayer state is simulated by the interface model.

5. There is no void between each layer of pavement structure, and the interlayer state is simulated by Interface model.

The interface has the characteristics of friction, cohesion, expansion, normal and shear stiffness, and tensile and shear bond strength, in which stiffness is a must parameter. In this numerical calculation, stiffness is used to describe the contact state between layers.

### 2.2.2. Reference Model

The numerical calculation in the following chapters takes the pavement structure of the Beijing–Harbin North Line in Hebei Province as the reference model, as shown in Figure 6. The main parameters of each pavement layer are shown in Table 2 below.

**Table 2.** Tensile and compression modulus ratio of asphalt mixture.

| Structure Layer | Thickness (cm) | Elasticity Modulus (MPa) |
|---|---|---|
| Upper layer | 4 | 9500 |
| Middle surface | 6 | 11,000 |
| base course | 10 | 9000 |
| The top layer of the subgrade | 18 | 11,500 |
| The middle layer of the subgrade | 18 | 11,500 |
| The lowest subgrade | 18 | 8500 |
| foundation | 200 | 187 |

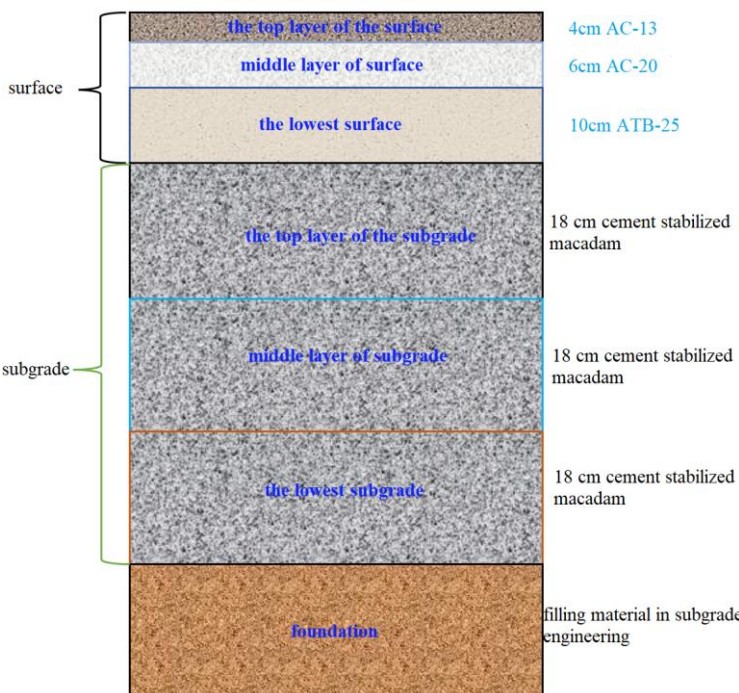

**Figure 6.** Pavement structure.

The vehicle's force on the road surface is transmitted through the contact between the road surface and the tire. In the pavement design calculation, the value of 100 kN double-wheel single-axle load group is used as the standard axle load; the single wheel and the pavement contact surface acting on the equivalent diameter is 21.3 cm, and the pressure is 0.7 MPa circular load. However, the tire–road interaction is complex, and the ground shape of the tire is not an ideal circle, but more like a rectangle or oval. In this paper, a rectangle is used to describe the vehicle load area.

In the existing pavement design theories, only vertical load is considered, while the effect of horizontal force is ignored. However, in the actual running process, the vehicle exerts both vertical and horizontal loads on the road surface. The horizontal force of the vehicle on the road surface is mainly affected by the vehicle running condition (uniform speed, normal acceleration, general braking, emergency braking) and the road type condition (plane, ramp road surface, turning, etc.). Generally, it is considered that the horizontal force F has a certain linear relationship with the vertical force P.

Although in the calculation of pavement structure, it is usually assumed that each pavement layer is infinite in the plane direction and the bottom layer is infinite in half space, it is not possible or necessary to take the model size as infinite in a numerical simulation calculation. Therefore, selecting the appropriate model size can not only ensure the accuracy of numerical analysis but also reduce the amount of calculation.

The grid division has a great influence on the computational efficiency and results: if the grid is too sparse, the calculation results cannot meet the accuracy requirements, and if the grid is too dense, the calculation amount will be increased. In this paper, the non-uniform mesh division method was adopted. The fine mesh division was adopted in all directions near the load, and the mesh gradually thins away from the load. Figure 7 shows the the benchmark pavement structure model, which was built with solid three-dimensional eight-node units. The pavement structure model had 93,774 nodes and 80,640 units. In numerical simulation, the initial velocity values of the side and bottom surfaces of the model were set as 0.

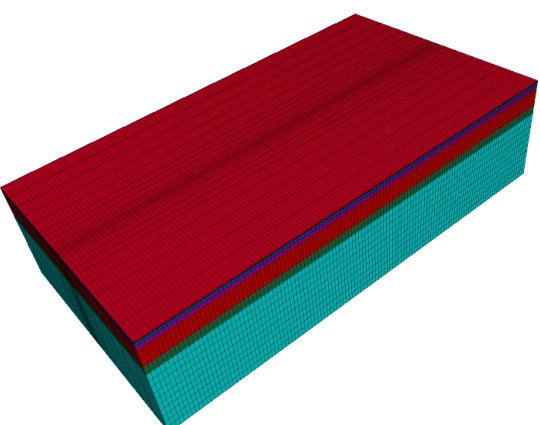

**Figure 7.** Reference model of the numerical simulation.

*2.3. Check Asphalt Pavement Simulation Model*

Before carrying out numerical simulation analysis, the numerical calculation results were compared with those in literature [18,19] when the middle layer was in a continuous state to verify the rationality and accuracy of the numerical model in this paper.

As can be seen from Figure 8, when the surface layer and the base layer were completely continuous, the tensile stress curves calculated by different methods had the same distribution trend and the numerical differences between them were small. When the continuous contact between the layers of the pavement structure was ideal, the whole pavement structure was under pressure and the stress of the road surface was at the maximum. Meanwhile, the pressure value decreased gradually along the depth. Within the depth of the surface layer, the structure was completely under pressure. The lower part of the base and the following part exhibited tensile stress. As can be seen from the calculation results in Figure 8, the simulation model adopted in this study had high computational reliability and was therefore used for the following calculation analysis.

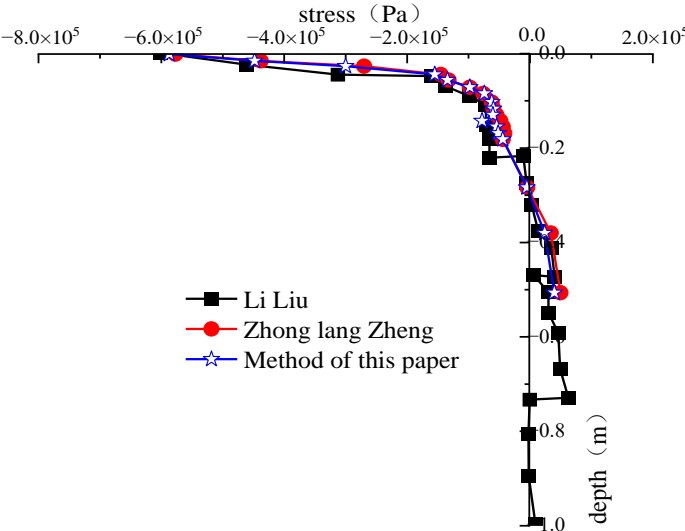

**Figure 8.** Reference model of numerical simulation.

## 3. Study on the Modulus Matching Mode of Asphalt Pavement Considering the Reflection Ability of Flexural and Tensile Fatigue Cracks

*3.1. Reflection Crack Simulation Based on MohrT and Interface Model*

The Mohr–Coulomb Tension Crack (MohrT) model can be used to simulate tension cracks by improving the traditional Mohr–Coulomb tension crack. Figure 9 below shows the stress nephogram obtained by the MohrT model and interface.

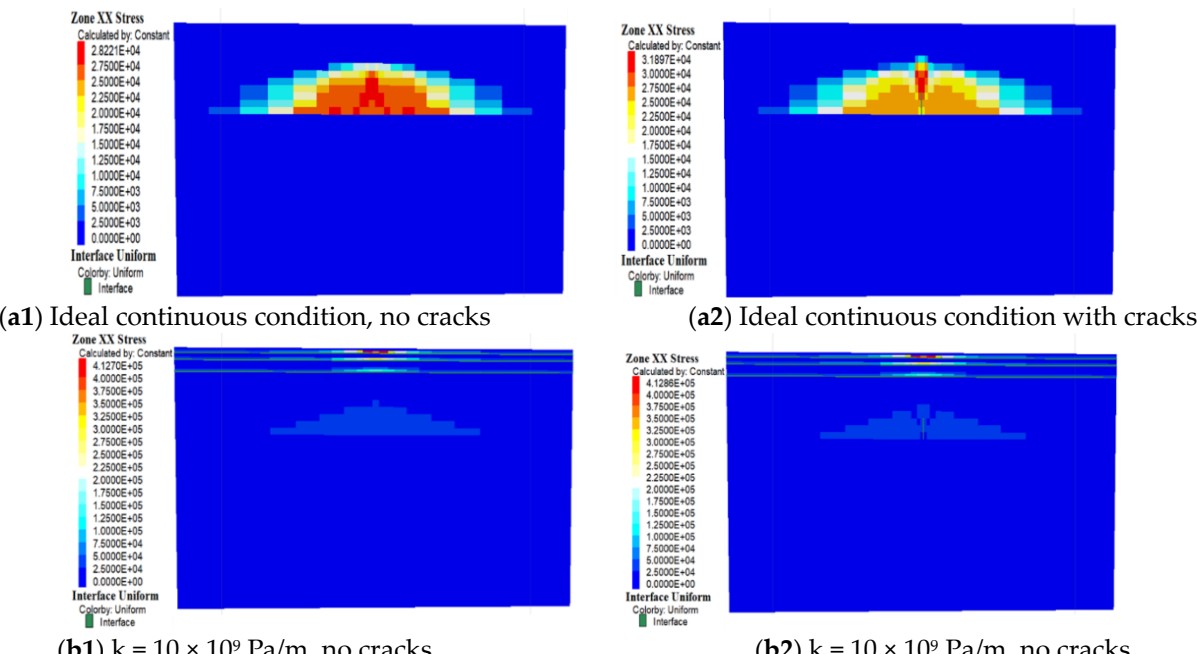

(**a1**) Ideal continuous condition, no cracks

(**a2**) Ideal continuous condition with cracks

(**b1**) k = 10 × 10⁹ Pa/m, no cracks

(**b2**) k = 10 × 10⁹ Pa/m, no cracks

**Figure 9.** Tensile stress nephograph of the pavement structure with andwithout cracks and different interlayer contact states.

Figure 9 shows the nephograph of tensile stress inside the pavement structure with and without cracks and different interlayer contact states in the reference model. As can be seen from Figure 9a, for the ideal continuous contact between layers, when cracks appear at the bottom of the base layer, the tensile stress in the base layer tended to increase and obvious stress concentration occurred near the crack tip. In addition, the results in Figure 9a also show that cracks in the base lead to an increase in the distribution range of tensile stress inside the base. Compared with Figure 9a, the difference in the shape of the stress nephogram in Figure 9(b1) and Figure 9(b2) is relatively small because the tensile stress is mainly concentrated in the surface layer when the contact between layers is weak.

Figure 10 shows the internal compressive stress increment (ideal continuous interlayer condition) caused by base cracks in the base model at different surface lamination tensile modulus ratios. As can be seen from Figure 10, the appearance of cracks in the base layer lead to an overall increase in the tensile stress inside the structure. Compared with the surface layer, the increasing trend in the base layer was more obvious, especially near the crack tip. Because the stress in the middle base changed from compressive stress when there was no cracking to tensile stress when there was cracking, the stress growth rate in the middle base was the largest. By comparing the stress growth rate of different compressive and tensile modulus ratios, it could be found that the influence of cracks on the surface increased with the increase of bending and tensile moduli.

Figure 11 shows the maximum tensile strain cloud image (ideal continuous interlayer condition) of the pavement structure (base and subgrade) with different moduli of the underlying layer when cracks occur in the underlying layer. It can be seen from Figure 11a that when cracks occurred in the base layer, the tensile strain mainly concentrated around the cracks, and its value reached its maximum at the tip of the cracks. The tensile strain decreased with the increase of the distance to the tip. In addition, by comparing the tensile stress nephogram with different moduli of the lower layer, it can be seen that the larger the modulus of the lower layer, the smaller the corresponding tensile strain. By comparing Figure 11a,b, it could be found that compared with the tensile strain when the lower layer increased to 15,000MPa, the tensile strain corresponding to the middle layer with the same high-modulus asphalt mixture was relatively larger.

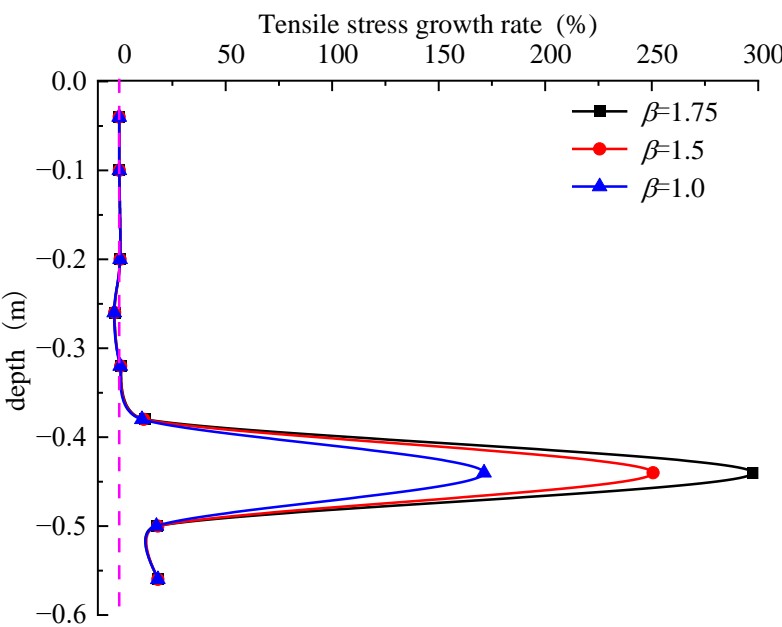

**Figure 10.** Influence of cracks on the tensile stress of the pavement structure under different compression and tensile modulus ratios.

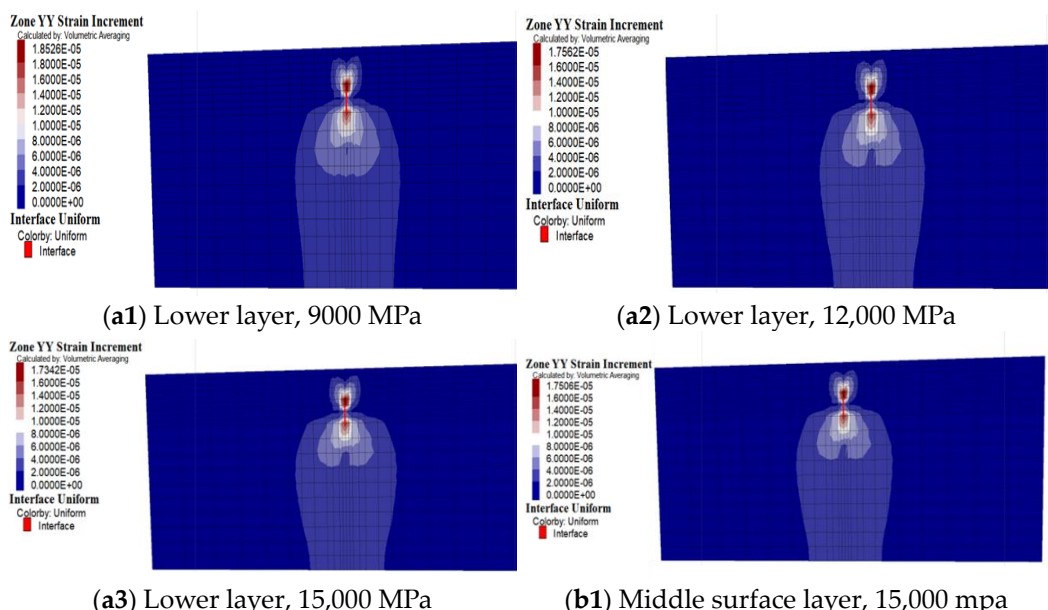

(**a1**) Lower layer, 9000 MPa

(**a2**) Lower layer, 12,000 MPa

(**a3**) Lower layer, 15,000 MPa

(**b1**) Middle surface layer, 15,000 mpa

**Figure 11.** Tensile strain cloud map of pavement structure when cracks occur in the base layer.

*3.2. Semi-Analytical Solution of the Stress Intensity Factor Based on Finite Element Simulation*

The stress intensity factor K is a key mechanical parameter representing the singularity strength of the stress field at the crack tip. Obtaining the stress intensity factor is the basis of evaluating the working performance and predicting the life of the seam structure. At present, the calculation of K is mainly divided into two categories: the numerical method and the weight function method. The weight function method was first proposed by Bueckner [20], which can quickly obtain stress intensity factors through mathematical integration and the superposition principle, avoiding complex mathematical modeling and

analysis processes. According to the definition of weight function, the stress intensity factor K can be expressed as follows:

$$
K = f(\alpha)\sigma_0\sqrt{\pi a}
$$
$$
f(\alpha) = \int_0^\alpha \frac{\sigma(\xi)}{\sigma_0}\frac{m(\alpha,\xi/\alpha)}{\sqrt{\pi a}}d\xi
\tag{16}
$$

where the type of $\alpha = (a/W)$, $\xi = (x/W)$ are, respectively, the crack length and position coordinates; $W$ is the geometric feature of crack; $m(\alpha,\xi/\alpha)$ is the weight function; $\sigma_0$ is the outer load; $\sigma(\xi)$ is without the stress of crack, which can be achieved by conventional finite element calculation results. The weight function method is the main function to determine $f(\alpha)$. According to the research of Rice, $f(\alpha)$, by any reference under the load of crack opening displacement $U_r(a, x)$, can be used to calculate

$$
m(\alpha,\xi/\alpha) = \frac{E'\sqrt{W}}{K_r}\frac{\partial U_r(a,x)}{\partial a}
$$
$$
K_r = f_r(\alpha)\sigma_0\sqrt{\pi a}
\tag{17}
$$

where $E'$ is the equivalent modulus and $K_r$ is the stress intensity factor under the reference load. For $U_r(a, x)$, scholars have proposed a variety of computational models. For example, Petroski and Achenbach [21] believe that it can be expressed as follows:

$$
U_r(a,x) = \frac{\sigma_0}{\frac{2E}{1-v^2}}\left(\frac{K_r}{\sigma_0(\pi a)^{0.5}}\frac{4a^{1.5}}{W}(a-x)^{0.5} + \frac{Ga^{0.5}}{W}(a-x)^{1.5}\right)
\tag{18}
$$

where $G$ can be obtained by substituting the abovementioned equation into Equation (16). Considering that the above formula only represents $U_r(a, x)$ as two terms, its accuracy may be low and its application may be limited; Wu [21] suggested using the following formula for calculation:

$$
U_r(a,x) = \frac{\sigma_0\alpha}{\frac{2E}{1-v^2}}\sum_{j=1}^{J}F_j(\alpha)[1-(\xi/\alpha)^2]^{j-0.5}
\tag{19}
$$

where Type, $F_j(\alpha)$ is the correction coefficient of the stress intensity factor and can be obtained through the relationship between the crack tip, which is self-consistent, and determined opening displacement conditions. According to Equation (19), the weight function can be expressed as follows:

$$
m(\alpha,\xi/\alpha) = \frac{\sum_{i=1}^{J+1}\beta_i(\alpha)}{\sqrt{\pi a}}[1-(\xi/\alpha)^2]^{i-1.5}
$$
$$
\beta_i(\alpha) = \frac{\alpha\frac{dF_{i-1}(\alpha)}{d\alpha}-(2i-4)F_{i-1}(\alpha)+(2i-1)F_i(\alpha)}{f_r(\alpha)}
\tag{20}
$$

where type (20) into type (16), and when there are no cracks at the same time, the stress $\sigma(\xi)$ is used to calculate the stress intensity factor K. If the stress on both sides of the crack is assumed to be linearly distributed, namely,

$$
\frac{\sigma(\xi)}{\sigma_0} = k|\xi| + b \qquad |\xi_1| \le |\xi| \le |\xi_2|
$$
$$
f(\alpha) = \frac{\alpha}{\pi}\left\{\sum_{i=1}^{J+1}\frac{\beta_i(\alpha)}{2i-1}[1-(\xi/\alpha)^2]^{i-0.5}\right\}_{\xi_1}^{\xi_2}
\tag{21}
$$

### 3.3. Verification of Semi-Analytical Solution of Stress Intensity Factor

In order to verify the rationality of the abovementioned method, it is compared with the calculated values of XFEM. Since the semi-analytic model in 12 relies on numerical simulation results, the mesh near the crack was encrypted to increase accuracy, and then the corresponding stress intensity factor was calculated.

Figure 12 shows the stress intensity factors obtained by the semi-analytic method and XFEM for different upper (lower) surface moduli. The results in Figure 12 show that the results obtained by the two calculation methods have a good consistency. When the upper (lower) surface modulus increased, the stress intensity factor showed a decreasing trend. When the upper (lower) surface modulus was greater than 14,000 MPa, the stress intensity factor decreased gradually. When the modulus value was relatively small, the influence degree of the middle layer and the lower layer's modulus on the stress intensity factor was similar, while with the increase of the modulus, the stress intensity factor corresponding to the middle layer's modulus was relatively larger. The abovementioned phenomenon indicates that, for the benchmark model in this study, the increase of the modulus of the lower layer can more effectively inhibit the propagation of reflection cracks at the base layer.

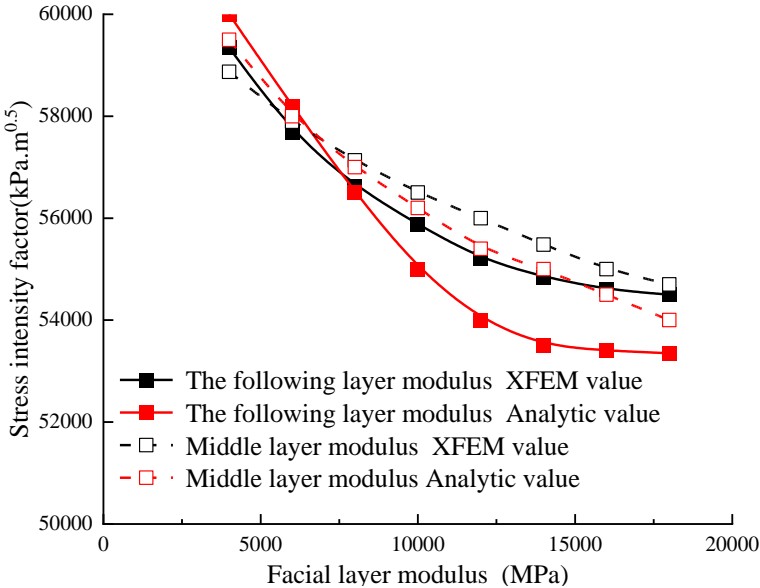

**Figure 12.** Variation curve of stress intensity factor with the modulus of the upper (lower) surface layer.

## 4. Study on the Influence of Surface Modulus on the Deformation of Pavement Structure Layer and the Compressive Strain of Subgrade Top Surface

### 4.1. Vertical Pavement Deformation

Figure 13 shows the vertical deformation nephogram of the cross-section of the reference model under the constraint state between different layers. As can be seen from Figure 13a, when the constraints between layers are poor, the vehicle load cannot be effectively transferred to the structure, and the strain is mainly concentrated in the upper and middle layers, resulting in the uneven and discontinuous distribution of vertical deformation within the structure. Different from the cloud image in Figure 13a, when the interlayer is in an ideal continuous condition, the vertical deformation is distributed more evenly and continuously between layers, that is, the overall working performance of the structure is better. In addition, by comparing the two cloud images in Figure 13, it can also be found that, compared with the structural deformation under ideal continuous conditions, the vertical deformation under poor interlayer constraints was larger.

Figure 14 shows the vertical deformation nephogram of the reference model with different underlying moduli under ideal continuous conditions. As can be seen from Figure 14a, when the underlying surface modulus is 6000 kPa, the vertical deformation of pavement under vehicle load is mainly concentrated just below the load, and the vertical deformation gradually decreases with the increase in the loading position. With the increase of the modulus of the lower layer, although the distribution characteristics of the vertical deformation nephogram did not change, the maximum vertical deformation value of the path table tended to decrease.

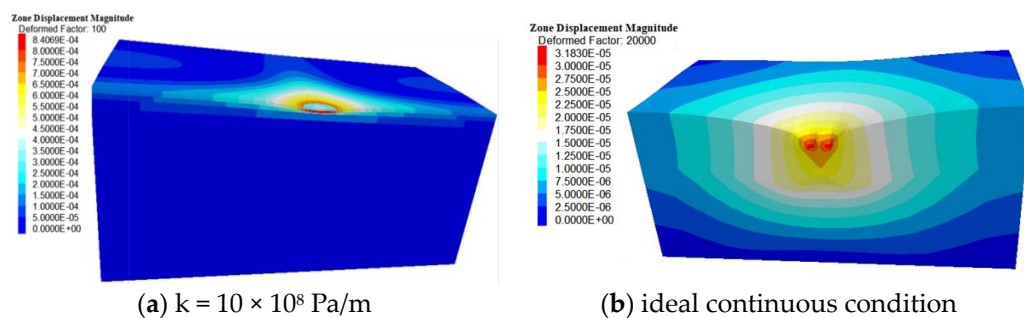

(**a**) k = 10 × 10⁸ Pa/m  (**b**) ideal continuous condition

**Figure 13.** Vertical deformation cloud of the reference model.

(**a**) 6000 kPa  (**b**) 12,000 kPa

(**c**) 15,000 kPa  (**d**) 18,000 kPa

(**e**) 21,000 kPa

**Figure 14.** Vertical deformation nephogram of the reference model under ideal continuous conditions with a different modulus of the underlying layer.

Figure 15 further shows the maximum vertical deformation curve of the path surface with different moduli of the middle and lower layers. As can be seen from Figure 15, when the interlayer constraint is poor and the stiffness is $10 \times 10^9$ Pa/m, the maximum vertical

deformation of the path surface presents an approximately linear decrease trend with the increase of the modulus of the middle or lower layer, and the decrease rate is greater under the condition of the modulus of the middle layer. It can be seen that when the modulus of the surface layer is less than 12,000 MPa, the vertical deformation of the road surface can be better reduced by placing it in the lower layer. To control the vertical deformation of the path surface, it can be placed in the middle surface layer when the modulus of the face layer is large. In addition, according to the abovementioned results, in the range of modulus variation shown in Figure 15, increasing the modulus can reduce the growth rate of vertical deformation. Different from the result when the interlayer stiffness is $10 \times 10^9$ Pa/m, when the interlayer is in an ideal continuous condition, although the vertical deformation of the path surface still decreases with the increase of the modulus, the vertical deformation value is smaller at this time. In the range of the overall modulus, the vertical deformation value of the lower layer modulus is smaller than that of the middle layer modulus condition. In addition, the nonlinearity of the growth rate of vertical deformation with the change of modulus is gradually enhanced, that is, with the increase of the modulus, its influence on the vertical deformation of the path surface is gradually weakened. The above results show that if the interlayer contact state of the pavement layer can meet the design requirements during the service period, the high-modulus asphalt mixture can be placed in the lower layer to achieve the effect of controlling the vertical deformation of the pavement surface.

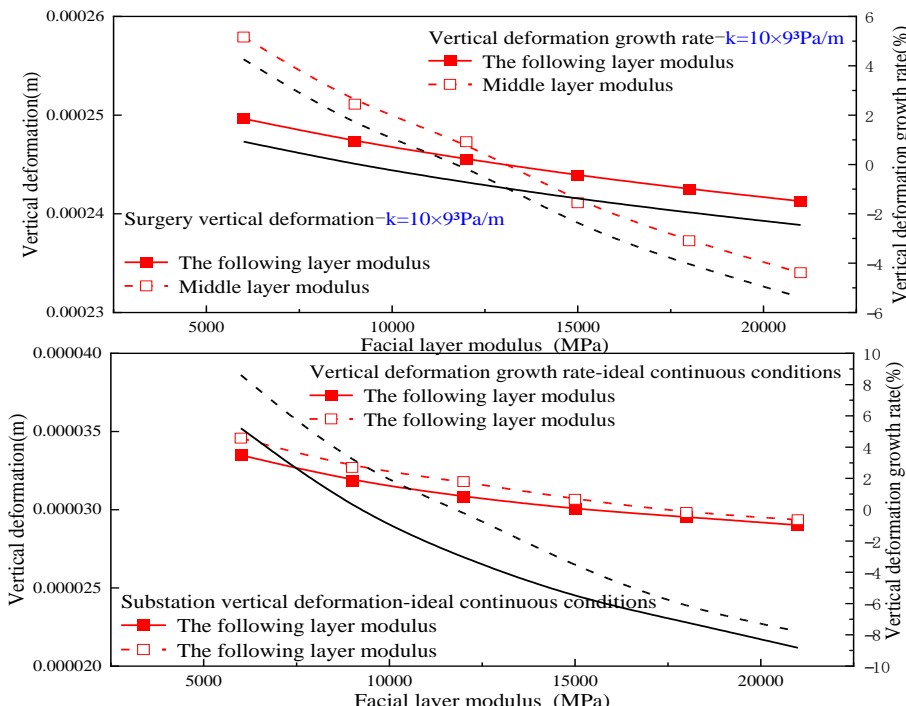

**Figure 15.** Curve of vertical deformation and growth rate with the modulus of the surface layer.

Figure 16 shows the variation of road surface settlement with the ratio of tensile modulus of surface lamination. The results in Figure 16 show that the vertical deformation of the road surface gradually increased with the increase of the tension modulus ratio of the surface lamination, but the overall change was not obvious, especially when β is greater than 1.5. In addition, similar to the results in Figure 15, the results in Figure 16 still show that the vertical deformation of the path table decreased with the enhancement of the constraints between layers. In addition, by comparing the relative sizes of vertical deformation between the two types of surfaces under the constraint state in Figure 15, it can be seen that the relative growth rate tended to decrease nonlinearly with the increase of the surface lamination tensile modulus ratio. The abovementioned phenomenon shows

that the influence of the constraint between layers on the vertical deformation of the path surface decreases with the decrease of the bending and tensile modulus of the surface.

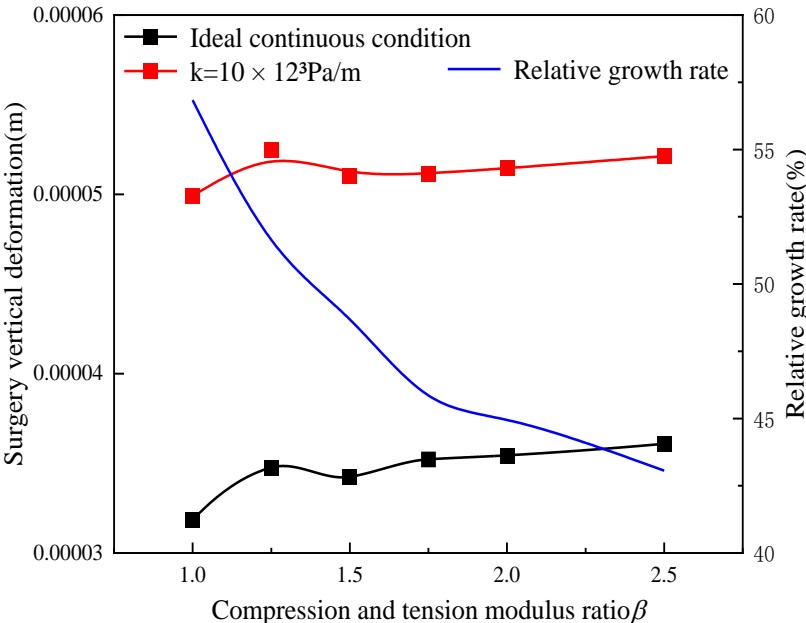

**Figure 16.** Curve of vertical deformation with tensile modulus ratio of surface lamination.

### 4.2. Subgrade Top Pressure Strain

Figure 17 shows the variation of subgrade top compressive strain with the ratio of surface lamination tensile modulus. The results in Figure 17a show that the compressive strain at the top of the subgrade under vehicle load presented a "pot bottom" type distribution, that is, the maximum compressive strain is under the load, and the nonlinear compressive strain decreased with the increase to the loading position. In addition, the results in Figure 17a also show that when cracks occurred in the bottom subgrade, the compressive strain at the top of the subgrade was further increased, especially near the cracks. With the strengthening of the constraints between layers, vehicle load can be transferred to the entire pavement structure more effectively, which leads to the overall increase of the compressive strain at the top of the subgrade with the increase of k.

Figure 18 further shows the variation law of maximum compressive strain at the top of the subgrade with the constraint stiffness k between layers. It can be seen from Figure 17 that with the increase of k, the maximum compressive strain at the top of the subgrade increased rapidly at first and then tended to be constant. That is, when the interlayer constraint strength reached a certain degree, it no longer had an impact on the compressive strain at the top of the subgrade. The reason for the abovementioned belief may be that, although more vehicle loads can be transferred to the road surface when the constraints between layers are increased, a relatively large compressive strain will be generated. However, with the further increase of the constraints between layers, the overall performance of the whole pavement structure is gradually enhanced, and the structural stress is more uniform so that the compressive strain of the top surface of the subgrade does not continue to increase. By comparing the relative size of the compressive strain between the subgrade top surface with and without cracks, it can be found that with the increase of k, the relative growth between the two increases first and then decreases and tends to a constant value.

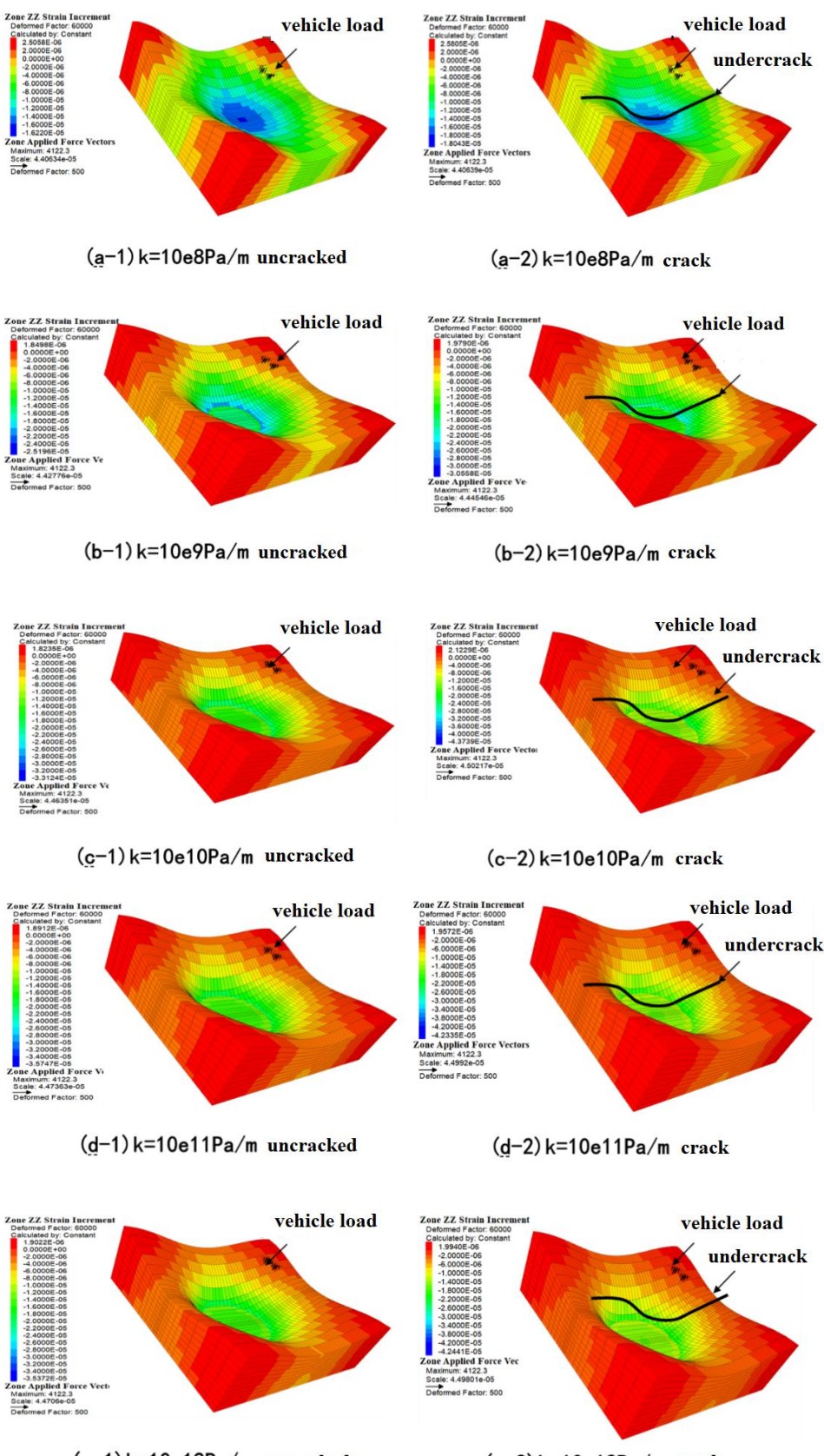

**Figure 17.** Pressure strain nebulae of the top surface of the subgrade of the reference model.

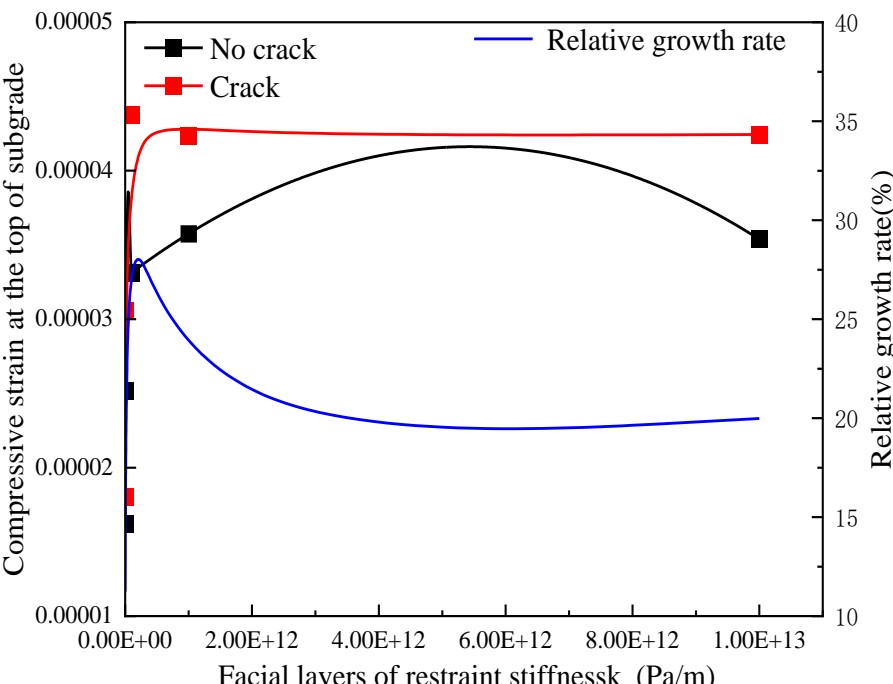

**Figure 18.** Variation curve of the maximum compressive strain of subgrade top with interlayer constraint stiffness.

Different Lamination Tensile Modulus Ratios

Figure 19 shows the pressure strain cloud map of the top surface of the base model when the interlayer stiffness k = 10 × 10^{10} Pa/m and the tension modulus ratio of different surface laminations $\beta$. The results in Figure 19 show that the compression-tension modulus ratio $\beta$ has little effect on the shape of the compression-strain nephogram. In addition, the compressive strain of the top surface of the subgrade gradually increased with the increase of the compression–tensile modulus ratio $\beta$. Figure 20 shows the variation of maximum compressive strain at the top of the subgrade with the lamination tensile modulus ratio $\beta$. According to Figure 20, when the interlayer constraint was relatively small, the maximum compressive strain at the top of the subgrade first increased and then tended to be constant with the increase of the compression–tensile modulus ratio. When $\beta$ is greater than 1.5, the maximum compressive strain at the top of the subgrade hardly changed. When the interlayer constraint was relatively large, the maximum compressive strain at the top of the subgrade increased with the increase of $\beta$ until $\beta$ = 1.75, and its influence on the maximum compressive strain at the top of the subgrade weakened when $\beta$ continued to increase.

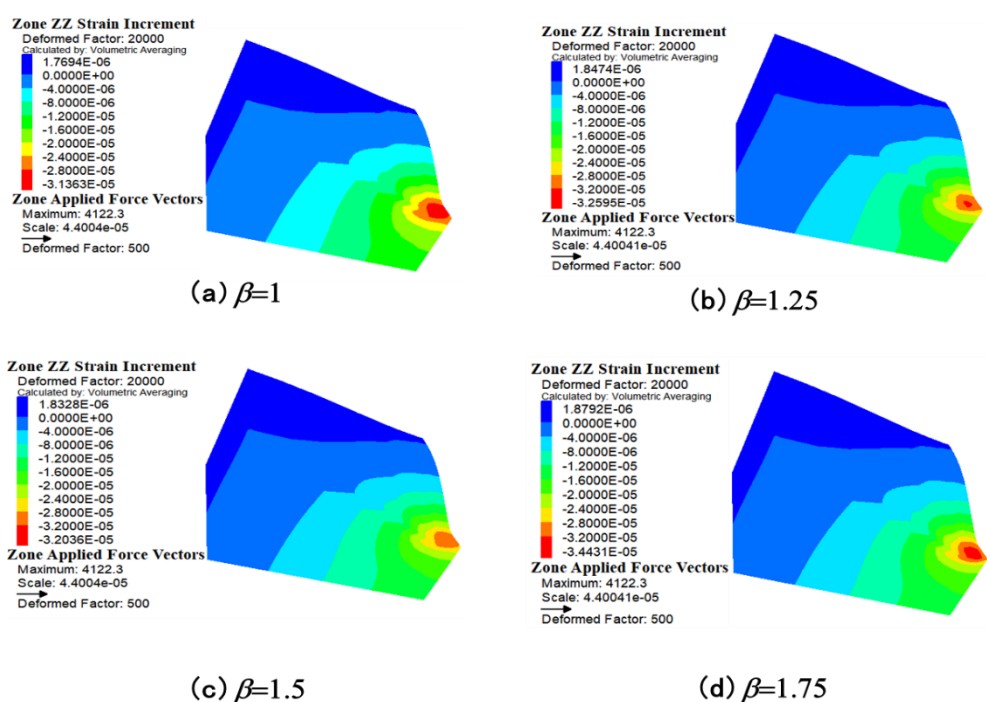

**Figure 19.** Compressive strain cloud map of the base model at the top of the subgrade with different lamination tensile modulus ratios.

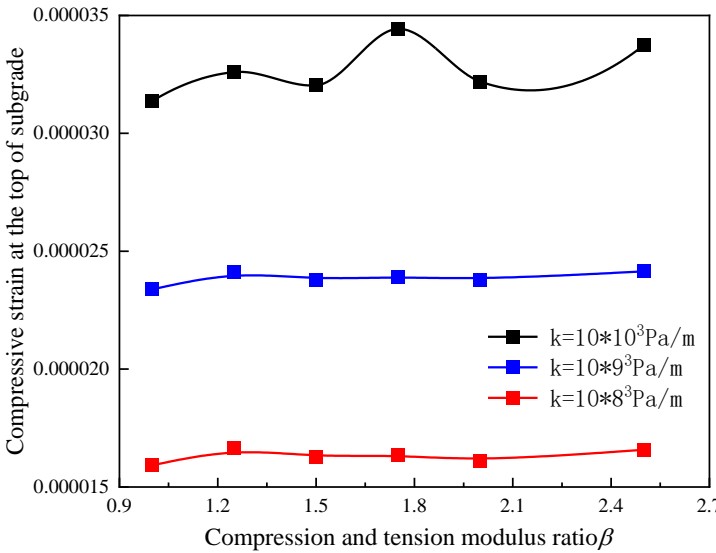

**Figure 20.** Variation curve of maximum compressive strain at the top of the subgrade with lamination tensile modulus ratio.

## 5. Conclusions

Based on the numerical simulation technology, the modulus matching mode of asphalt pavement is studied, and the reflection ability to bend and tensile fatigue crack, the vertical deformation of the pavement structure layer, and the compressive strain of the subgrade top surface are analyzed. The main conclusions are as follows:

(1) The appearance of cracks in the base will lead to obvious stress concentration at the crack tip, and the tension area in the base will increase. Considering that the interlayer constraint of the pavement structure is not an ideal continuous condition during the service period, it leads to greater tensile stress or strain at the bottom of the upper layer. Therefore, in order to avoid cracking of pavement structure, high-modulus



asphalt mixture can easily be used in the lower layer, and the tensile strength of the bottom of the surface layer should be enhanced or the upper layer should have a small compression modulus ratio;

(2) When the contact state between layers of pavement is good, the lower layer with a high modulus can be used to reduce the deformation of the path surface; When the contact state between layers is poor and the modulus of the asphalt mixture is large, it can be placed on the middle surface layer to control the vertical deformation of the road surface;

(3) With the decrease of the modulus of the bottom layer and the increase of the tensile modulus ratio of the surface laminating, the influence of the base crack on the tensile stress and strain is more obvious. With the increase of the surface modulus, the stress intensity factor determined by the semi-analytical method shows a nonlinear decrease trend, and the change in stress intensity factor is not obvious when the modulus is greater than 10,000;

(4) Under the action of vehicle load, the compressive strain of the top surface of the subgrade presents a "pot bottom" type distribution, and the maximum compressive strain is relatively larger when cracks occur at the base. The vertical deformation of the road surface and the compressive strain of the subgrade top surface gradually increase with the increase of the tensile modulus ratio of surface laminate, especially when the compressive tensile modulus ratio is greater than 1.5. In addition, the effect of the constraint between layers on the vertical deformation of the path surface decreases with the decrease of the bending and tensile modulus of the surface;

(5) Considering that the interlayer constraint of pavement structure during the service period is not an ideal continuous condition, which leads to greater tensile stress or strain at the bottom of the upper layer. Therefore, in order to avoid cracking of pavement structure, a high-modulus asphalt mixture can easily be used in the lower layer, and the tensile strength of the bottom of the surface layer should be enhanced or the upper layer should have a small compression modulus ratio.

**Author Contributions:** Conceptualization, G.Y. and P.X.; methodology, P.X.; software, X.L.; validation, G.Y., Y.H. and C.P.; formal analysis, H.M.; investigation, G.Y.; resources, H.M.; data curation, C.P.; writing—original draft preparation, G.Y.; writing—review and editing, X.L.; visualization, X.L.; supervision, X.L.; project administration, Y.H.; funding acquisition, Y.H. All authors have read and agreed to the published version of the manuscript.

**Funding:** This research was funded by Science and Technology Project of Hebei Provincial Department of Transportation (TH1-202017).

**Institutional Review Board Statement:** Not applicable.

**Informed Consent Statement:** Not applicable.

**Data Availability Statement:** Not applicable.

**Conflicts of Interest:** The authors declare no conflict of interest.

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
