# Peer review of "Performance-Based Expressway Asphalt Pavement Structural Surface Layer Modulus Matching Mode Research"

_coatings, doi:10.3390/coatings13060986_

Round 1
Reviewer 1 Report
This paper presents verification of applicability of different modular matching modes supporting the design of the asphalt pavement structure, based on structural computing and simulations. Authors provide very interesting investigation and present among others: the reflection ability of bending and tensile fatigue cracks, the vertical deformation of pavement structure layer, and the compressive strain of subgrade top surface.
Although presented article is very well written and may be interesting to the readership of this journal, the paper may only be considered for publication after the following concerns have been addressed successfully in a minor revision:
1) The Introduction part needs some improvement. It should introduce the Readers to the subject and problems related to the research on one hand, and show what should be done on this field on the other. Now, in my opinion, this condition is not entirely fulfilled. I couldn't find the convincing evidence of the innovative nature of current research. What is a novelty in this article compared to others of this type? Please make it visible what was the purpose of the research and what made them innovative or important. I know that question is obvious, but you need to introduce the Readers to the subject and prove or set up the thesis. Both of these things have not even properly outlined here.
2) Authors should read the whole manuscript once again carefully and correct some strange signs appearing in the text (for example "@" can be found in line 146 and 177).
3) Figures should be larger due to the large amount of information (graphs) they contain. Especially the labels related to colours (like in fig. 17) are barely legible. Also the units (like in fig. 15 and 16) should be rather scientific than basic type, which helps reducing the number of zeros in the notation.
4) In line 177 it should be rather "Figure 4-5" than "Figure 2-3". Am I right?
The whole text in my opinion should be readed once again carefully and "polished" due to some stylistic mistakes. These are not big problems, but it should be definitively corrected before publishing.
Author Response
Thank you again for your Suggestion. I made the following modification and explanation.

Reviewer 2 Report
Dear authors, your article has a scientific novelty and good practical value. In order to be published I propose to do some corrections.
1. In the Abstract section, it is necessary to present some numerical results of the research, as well as prospects for further research and implementation of the results obtained.
2. It is necessary to unify the form of designation of links. References 1, 2, 3 (for example, lines 46, 53 and 58) and Cheng Huilei et al. (2022) (for example, line 64) are present in the work.
3. In Fig. 5 there are 2 drawings that do not have a designation.
4. The captures of figures and subsections must begin with a capital letter.
5. Fig. 7 is not very clear, it should be removed.
6. Fig. 9 is better to correct, namely to remove empty spaces.
7. A space is required between the authors and the reference number, such as lines 317 and 320.
8. Fig. 14 and 17 should be on one page (a, b, c, d, e)
9. Fig. 15 shows 2 graphs under the same name.
10. It is necessary to develop mathematical dependencies that describe patterns on charts
11. References 9, 20, 21, and 22 can be deleted or replaced by newer ones.
Author Response

(The authors gave the same response as above.)

Round 2
Reviewer 2 Report
Dear authors, you have corrected the article and in this form, this paper can be published.